# How does the brain combine generative models and direct discriminative computations in high-level vision?

## Scientific question

Our question is how the primate brain combines generative models and direct discriminative computations in high-level vision. Both approaches aim at inferring behaviorally relevant latent variables $\mathbf{y}$ from visual data $\mathbf{x}$. In a probabilistic setting, the inference of the posterior $p(\mathbf{y}|\mathbf{x})$ is known as discriminative inference. The two approaches differ in how discriminative inference is implemented. In the generative approach, a model of the joint distribution $p(\mathbf{y}, \mathbf{x})$ of the latent variables and the visual input is employed. This model captures information about the processes in the world that give rise to the sensory data. Approximate inference algorithms are then used to infer

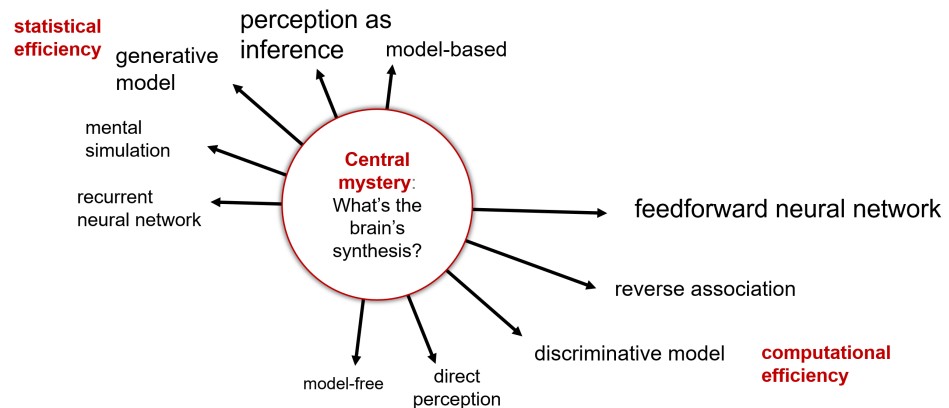

**Figure 1: Two contrasting visions of vision.** Separate traditions rooted in an empiricist and a rationalist conception, respectively, have envisioned the computations underlying visual perception as either a largely bottom-up, feedforward process of extracting behaviorally relevant information or as an inference process that interrogates the sensory evidence in light of a generative model that captures prior knowledge about the processes in the world that give rise to the sensory data. Several theoretically independent dimensions (double arrows) are conflated when considering the two perspectives as a dichotomy. The goal of this GAC is to disentangle the dimensions and clarify how they relate to each other, to understand how the favoured algorithm depends on the visual task, and to develop experiments that will help us understand how the primate brain combines elements of both conceptions.

the posterior over the latents given an image by estimating $p(\mathbf{y}|\mathbf{x}) = p(\mathbf{y}, \mathbf{x})/p(\mathbf{x})$. In the direct discriminative approach, a direct mapping from the sensory data to the posterior over the latents $p(\mathbf{y}|\mathbf{x})$ is learned without the use of an explicit generative model. The generative approach enables unsupervised learning of the structure of the world and promises better generalization to novel situations (statistical efficiency). Direct discriminative computations promise faster inferences (computational efficiency) that are accurate for new samples from the distribution experienced in training. In practice, inference of the full posterior may not be realistic and the visual system may settle for point estimates in certain cases.

## Background

Two contrasting conceptions have driven research on biological vision as well as the engineering of machine vision. The first emphasizes bottom-up signal flow, describing vision as a largely feedforward process that filters and transforms the visual information, so as to remove irrelevant variation and represent behaviorally relevant information in a format suitable for downstream functions of cognition and behavioral control. In this empiricist conception, vision is driven by the sensory data and perception is direct [1] in the sense that the processing proceeds from the data to the latent variables of interest. The alternative conception is that of vision as an inference process [2], where the sensory evidence is evaluated in the context of a generative model that captures prior knowledge about the world (ideally probabilistically). In this rationalist

conception, vision appears as an interrogation of the sensory evidence in a process often thought to involve top-down predictions of sensory data data serving to evaluate the likelihood of alternative hypotheses [3, 4]. The two perspectives are opposite in many respects and have deep roots in different philosophical traditions (empiricism, rationalism). Recent decades have brought major advances with both approaches in computer science and statistics, which have emphasized, respectively, computational and statistical efficiency. Biological brains appear, mysteriously, to combine the advantages of the two modes of visual inference, achieving both rapid recognition and robust inferences on the basis of limited and novel sensory data. Could it be that our intellectual heritage unduly polarizes our intuitions about the algorithm of vision, holding us hostage in a false dichotomy?

## Challenge or controversy

Both approaches have a long history and have achieved substantial successes in both computer vision [5, 6] and the computational neuroscience of vision [4, 7]. The generative approach is consistent with the notion that the primate brain performs mental simulations of processes in the world (at some level of abstract representation) [2, 8, 9]. The discriminative approach is thought to involve feedforward computations in a hierarchy of representations, as implemented in deep neural networks [10, 11]. It seems clear that the brain leverages the advantages of both modes of visual computation, and researchers have begun to combine these modes of inference in computational models [12–15]. The controversy therefore is not binary (Does vision rely on an inference process that inverts a generative model or on direct discriminative computations?), but rather concerns how primate vision combines elements of both approaches in the context of particular visual tasks.

A hodgepodge of related concepts from different fields (cognitive science, artificial intelligence, neuroscience, statistics), whose precise definition and interrelationships remain murky, currently hampers progress toward a nuanced perspective on how the primate brain may elegantly combine these modes of inference. The lack of a clearly defined shared language and a unifying theoretical framework may perpetuate a false dichotomy, suggesting that the algorithm of primate high-level vision is either purely generative (model-based) or direct discriminative (model-free), and keeping researchers polarized in contrasting intuitions.

## Competing hypotheses and proposed approach for resolution

In this GAC, we aim to make *theoretical* and *empirical* progress toward a deeper understanding of how the primate brain combines elements of both conceptions or achieves an algorithmic synthesis in which the two modes appear as extreme special cases. Theoretically, we aim to clarify the concepts and their relationships and to consider how different algorithmic solutions may be favourable for different tasks under particular resource constraints. Empirically, we aim to agree on experimental plans that enable us to localize the algorithm employed by a subject in the multidimensional space of algorithmic solutions.

**Survey:** We will gauge the GAC team members' and the CCN community's perspectives by administering a survey in which participants judge a collection of 20 to 40 propositions about model-based and model-free visual inference. We will administer the survey in the context of the GAC workshop, so as to identify the most contentious particular issues at the outset. We will repeat the survey in the middle and at the end of the GAC project, so as to track the evolution of perspectives among the GAC team members and the CCN community.

**Goal 1: Conceptual clarification.** The first theoretical goal of this GAC is to overcome the false dichotomy and to disentangle the different dimensions often conflated when considering model-based and model-free vision (including at least the following and possibly additional dimensions: discriminative / generative, mental simulation / reverse association, feedforward / recurrent; Fig. 1). The GAC will develop a common language that clarifies the distinctions and relationships among the relevant concepts from cognitive science, neuroscience, statistics, and machine learning. We will begin with the assumption that the ideal-

ized extremes of model-based and model-free vision are diametrically opposed corners of a multidimensional hypercube of algorithmic solutions to visual inference problems. We will then clarify to what extent the dimensions are independent or related, which combinations have precedents in the modeling literature, and which, if any, are theoretically impossible or nonsensical.

**Goal 2: Predictions from a resource-rational perspective.** The rationalist perspective motivates the generative approach to visual inference, which makes optimal use of limited data, but has long been understood to be unrealistic given constraints on computational resources [16]. We will consider the question of generative and discriminative visual computations from a resource-rational perspective [17–21]. The specific combination of generative and discriminative computations that an agent uses in the context of a particular task may depend on the agent's statistical and computational constraints. Statistical constraints are imposed by the amounts of evidence accessible to the agent for perceptual inference and learning. Computational constraints include limits on the time and energy for computation and on the computing components that fit into the skull. In addition, task properties (e.g., the complexity of the input-output relationship or the temporal requirements) may make a task more amenable to generative inference or direct discriminative computations. We plan to develop a resource-rational theoretical framework, in which the combination of generative and discriminative computations arises as a consequence of task demands and resource constraints on inference.

**Goal 3: Critical experiments for high-level vision.** The question of how to combine model-based and model-free modes of inference arises in several contexts, notably in the context of reinforcement learning. This GAC focuses on high-level vision. The co-organizers have complementary expertise (e.g., object recognition, face perception, dynamic object-vision, physical scene understanding, visual memory, in human and nonhuman primates). We will design experiments that identify where in the multidimensional space of algorithms primate high-level visual inference falls in a particular task. The resource-rational perspective (Goal 2) predicts that the algorithm will depend on the computational and statistical efficiency demands. Tasks that are typically employed in a particular sub-domain of high-level vision (e.g., in object recognition or scene understanding) may lend themselves to a particular combination of model-free and model-based computations, possibly slanting the consensus in that sub-domain towards one or the other. In the spirit of adversarial collaboration, novel tasks will therefore aim to challenge this consensus. These tasks will then be used in experiments in the participating labs while acquiring both neural and behavioral data. We aim to present preliminary results of these investigations and discuss their implications for the resource-rational perspective at CCN 2022.

## Concrete outcomes

(1) A common language that clarifies and relates relevant concepts from cognitive science, neuroscience, statistics, and machine learning, and defines the dimensions that span the hypothesis space of algorithms.
(2) A resource-rational perspective on the use of generative inference and direct discriminative computations for high-level vision.
(3) A set of critical experiments for high-level vision that will be pursued in the extended group of this GAC. Preliminary results will be presented at CCN 2022.

## Benefit to the community

The proposed GAC brings several benefits to the CCN community. First, the conceptual clarification and negotiation of a common language will facilitate scientific communication and foster collaboration among the cognitive, computational, and neuroscience subcommunities. Second, the resource-rational unifying theoretical perspective on generative and discriminative computations in high-level vision will stimulate new research within and beyond the field of high-level vision over the next 5-10 years. Finally, the GAC addresses a fundamental question of primate visual perception. The theoretical and empirical results will have broad implications for other systems of the primate brain and for AI engineering.

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

## Core group

The organizers, though individually polarized, span a range of perspectives and are committed to make progress toward a clearer theoretical articulation of the challenge and a balanced empirical assessment. This GAC aims for a larger number of perspectives, so as to be able to negotiate a shared language that avoids conflation of separate concepts and has the potential to become widely accepted. One goal of the kickoff event is to include a number of additional, particularly postdoctoral, researchers, and to assign core and senior advisor roles to all members. The members initial roles are to articulate their unique perspective on the role of generative models and discriminative inference in high-level vision.

- **James J. DiCarlo** has been a major proponent of the feedforward discriminative approach to explaining core object recognition, and his lab has recently explored the power of recurrent computations.

- **Ralf Haefner** is a proponent of trying to understand visual processing as probabilistic inference on a generative model with neural sampling as the brain's approximate inference algorithm.

- **Leyla Isik** has a strong background modeling visual computations, including the recognition of dynamic actions, with feedforward neural networks.

- **Talia Konkle** has studied the spatial organization of the human ventral stream and used both discriminitive models and contrastive unsupervised models to account for high-level visual representations.

- **Nikolaus Kriegeskorte** has used feedforward and recurrent neural networks with discriminative training with the goal to engage, from the bottom up, the generative elements of the inference process.

- **Benjamin Peters** uses recurrent neural networks to understand the contributions of direct discriminative and generative inference in human dynamic object vision.

- **Nicole Rust** has a strong background investigating the primate ventral stream from a discriminative perspective with a recent focus on the influence of memories of past experiences.

- **Kim Stachenfeld** has a background studying learned representations that support efficient model-free and model-based reasoning.

- **Josh Tenenbaum** has been a major proponent of the model-based inference approach in explaining human dynamic object vision and physical reasoning.

- **Doris Tsao** is a leading expert in primate face perception, studying both generative and discriminative algorithms as models of primate face perception.

- **Ilker Yildirim** has a strong background in using generative models as models for human perception and has recently combined generative models with efficient discriminative inference.

## Statement of commitment

We commit to to the GAC process, including:

- Incorporating feedback from the community and potentially welcoming new CCN community members to the GAC based on their written commentary to the GAC proposal

- Running an online kickoff workshop for CCN2021, inclusive of both founding core GAC members and those new members who joined through the community feedback process

- Writing the position paper to be submitted early 2022 to a curated special issue of NBDT, to be accompanied by commentary pieces authored by attendees of the CCN2021 kickoff workshop

- Attending and presenting progress at the following CCN2022

**Signed**:

James J. DiCarlo, Ralf Haefner, Leyla Isik, Talia Konkle, Nikolaus Kriegeskorte, Benjamin Peters, Nicole Rust, Kim Stachenfeld, Josh Tenenbaum, Doris Tsao, Ilker Yildirim

