# OpenReview forum: "How does the brain combine generative models and direct discriminative computations in high-level vision?"
_ccneuro.org/CCN/2021/Workshop/GAC_

### Official Review · ~Adrien_Doerig1 · 2021-07-22
**Important question, interesting approach, sparse experimental description**

**Rating:** 8
**Confidence:** 4

**Review:**

DiCarlo et al. propose to confront two important conceptions about the visual system: a generative approach, in which the visual system is seen as using a model of the world and how it gives rise to sensory data, versus a discriminative, in which the visual system is seen as implementing a direct mapping from the data to the result of perceptual inference, without using a generative model. In sum, the authors ask to what extent the visual system uses a generative model.

To this end, they propose, first, to conceptually clarify the issue. I agree with the authors that this will be beneficial for all the different fields concerned by the question. A common language to tackle these issues is indeed needed.

Second, they propose to develop a theoretical framework: the resource-rational perspective. They suggest that the generative/discriminative dichotomy may be a false one, and that the brain may rely more on one or the other approach depending on the situation. This proposal sounds promising to me.

Third, they propose to design and conduct experiments studying the generative/discriminative tradeoffs in high level vision across a number of tasks, and test the predictions of the resource-rational framework. I agree of course that experiments are needed. However, there is not enough information in the proposal to tell which experiments will be carried out, or how.

Overall, I think this is a timely and important topic to study. And I recommend accepting it.

One limitation is that I do not really see this proposal as an *adversarial* collaboration, since all authors seem to agree that the resource-rational perspective is the way to go. There is a slight concern that this may make the collaboration less adversarial and therefore a little off topic for this "venue" and perhaps a little less effective. However, I am not very familiar with this GAC format yet, so I may be wrong.

Another limitation is that it is hard to judge the experimental approach based on this proposal since no details, nor even a sketch of which ideas will be employed, are provided. The authors are all very well regarded in the field and can be trusted in this respect, but the proposal is still weak in this regard. Again, being unfamiliar with GAC proposals, I may be wrong in assuming that a detailed description is reuired.

A last minor comment is that I am not sure that the term "rationalist" is adequate for the "generative" approach. At least it is not the same sense as is usually given to the word "rationalist". Indeed, there is no "reasoning" involved.

In summary, this is a very relevant topic with a sensible theoretical approach. Even though there does not seem to be a real adversarial setup, nor a precise experimental description in this proposal, I recommend to accept.

---

### Official Review · ~David_G_Nagy1 · 2021-07-26
**Interesting GAC**

**Rating:** 8
**Confidence:** 4

**Review:**

This is a joint review by Gergo Orban, David Nagy, Balazs Meszena and Marton Hajnal.

The proposal contrasts two dominant approaches to understanding high-level vision: the discriminative approach where the brain learns to infer latent variables directly, thus learning a conditional distribution of latents given observations, and the generative approach where the joint distribution of latents and observations is learned and the generative model is inverted to compute the latent variables.

We agree with the authors that viewing these perspectives as a dichotomy conflates dimensions which can be misleading as to what specific hypothesis is being tested in a given experiment. Consequently, we also agree that it is a worthwhile goal to invest into developing terminology that clarifies the dimensions along which high-level vision can be described. We would highlight a few points that seem particularly interesting to focus on.

One important point would be to have a precise definition of what is meant by the two contrasted approaches. Both approaches agree that an important goal of the visual system is to compute p(y|x). One interpretation of the distinction between the approaches is on the algorithmic level: is the computation of the posterior p(y|x) implemented such that an explicitly represented generative model is inverted? A second interpretation remains on the computational level: does the visual system in some sense contain an explicitly represented generative model in addition to a discriminative model?

The proposal points out that the generative approach enables unsupervised learning, and we feel it would be interesting to discuss to what extent the approaches map to unsupervised vs supervised learning. Specifically, does “directly learning p(y|x)” assume direct access to x,y pairings so that the mapping between them can be directly approximated i.e. can the discriminative approach be equated to supervised learning? Looking at the two competing frameworks from a supervised/unsupervised point of view highlights an additional axis along which one needs to explore interpolations: weakly supervised and active learning paradigms can have critical contributions to the emerging representations and therefore shape how the computational architecture can be identified.

The authors identify statistical efficiency and computational efficiency as distinct characteristics of the competing approaches. Recent advances in probabilistic models proposed amortised inference, which can make the inversion of the generative model more efficient over the course of learning (e.g. Dasgupta & Gershman, 2021). It could be worthwhile to clarify where the resulting inference algorithm lies in the proposed framework - would it still count as inverting the generative model? Is the resulting algorithm distinguishable from using a discriminative approach?

The proposal introduces the terminology of model-based vs model-free vision. While we see an intuitive appeal in making this analogy, we would like to point out that a precise classification of algorithms into MF or MB and mapping them to cognitive processes or regions in the brain has been fraught with many issues, as recently reviewed by Collins & Cockburn, 2020, which might serve as a good jumping off point.

The proposal points out that both approaches (discriminative and generative) infer behaviorally relevant latent variables y and that the main difference is how this inference is implemented. We note that the sets of these latents are often different in the two approaches. In the discriminative case y typically consist only of the highest level features (labels), while the generative approach includes a larger set of latent variables.

In summary, the proposal is well-articulated and touches on very timely topics. It will be exciting for the wider community and can establish communication between theory of computations and experimentalists.

---

### Official Review · ~Gido_Martijn_van_de_Ven1 · 2021-07-26
**Suggestion to consider incremental learning settings**

**Rating:** 8
**Confidence:** 4

**Review:**

This proposal addresses a classic question in computational cognitive neuroscience: when the brain performs inference (e.g., classification of visual objects), does it use a discriminative approach (i.e., it infers p(**y**|**x**) directly) or a generative approach (i.e., it infers p(**y**|**x**) indirectly through Bayes’ rule). The proposal expresses the viewpoint that rather than a binary choice, this question should be phrased more nuanced and be about how the brain combines discriminative and generative computations. The proposal has three main goals: (1) to develop a common language and clarify concepts in order to facilitate interaction between different fields (e.g., cognitive science, neuroscience, machine learning, statistics) in which there is work relevant to this topic; (2) to develop a theoretical framework based on both resource and rationalist considerations for how the brain combines generative and discriminative computations in different situations; and (3) to design experiments to test the predictions of this framework (or to inform its development).

I think this is a strong proposal about a topic well suited for a GAC project. I believe the effort to develop a shared terminology across different fields could be extremely valuable, and I think the authors have already started to make good progress towards this goal (e.g., Figure 1). I am also excited about the plans to develop a resource-rational theoretical framework and to experimentally test its predictions. However, the proposal is rather general with regards to these two goals, and it would strengthen the proposal to make them more concrete.
Nevertheless, I think this is a good proposal and I hope it will be accepted. In this review I mainly want to make a suggestion, which could help to make the resource-rational framework more concrete.

My suggestion for this proposal is to include consideration of incremental learning settings in which the latent variable **y** changes over time, as I believe this setting would provide a valuable new perspective to the central question of this proposal.
To illustrate this, let’s consider an example of such an incremental learning setting: class-incremental learning. In this problem, an algorithm must learn to distinguish between classes that are encountered incrementally (e.g., first all examples from class 1 are observed, then all examples from class 2). Because not all classes are observed together, directly learning a discriminative classifier is very challenging: it is difficult to find a rule to distinguish between two classes when directly comparing between them is not possible. One way around this is to use ‘replay’ (e.g., Shin et al., 2017 *NeurIPS*, https://arxiv.org/abs/1705.08690): that is, when training on examples from a new class, samples representative of past classes are interleaved. This way, it is still possible to learn a discriminative classifier. Importantly, to enable such replay, a generative model of the previously seen classes must be learned. (Note that, for the sake of this argument, storing [a subset of] examples from past classes can also be seen as learning a generative model.) An alternative solution for class-incremental learning is, rather than learning a discriminative classifier, to learn a generative classifier (van de Ven et al., 2021 *CVPR workshop*, https://arxiv.org/abs/2104.10093).
Therefore, in the incremental learning setting, the important question might not be whether a generative model is used, but how it is used: The generative model can be used either directly to perform generative inference, or it can be used indirectly for generating samples to train a discriminative model on.